# Grape (*Vitis vinifera* L. cv. País) Juices Obtained by Steam Extraction

Walther Ide [1], Constanza Sabando [1], Johanna Castaño [2], Natalia Pettinelli [1], Richard Bustos [3], Ana Linares [1], Leandro Mora [3], Niels Müller [4], Guillermo Pascual [3,*] and Saddys Rodríguez-Llamazares [1,*]

1 Centro de Investigación de Polímeros Avanzados, Edificio Laboratorio CIPA, Avenida Collao 1202, Concepción 4081112, Chile; w.ide@cipachile.cl (W.I.); c.sabando@cipachile.cl (C.S.); n.pettinelli@cipachile.cl (N.P.); linaresartal@gmail.com (A.L.)
2 Facultad de Ingeniería y Tecnología, Universidad San Sebastián, Lientur 1457, Concepción 4080871, Chile; Johanna.castano@uss.cl
3 Facultad de Agronomía, Universidad de Concepción, Edmundo Larenas 64, Concepción 4070386, Chile; ricbustos@udec.cl (R.B.); lemora2016@udec.cl (L.M.)
4 Unidad de Desarrollo Tecnológico, Universidad de Concepción, Avda. Cordillera 2634, Coronel 4191996, Chile; n.muller@udt.cl
* Correspondence: gpascual@udec.cl (G.P.); s.rodriguez@cipachile.cl (S.R.-L.); Tel.: +56-41-220-4000 (G.P.); +56-41-3111852 (S.R.-L.)

**Abstract:** *Vitis vinifera* L. cv. País is an ancestral Chilean grape undervalued due to its undesirable oenological characteristics. In this study, steam extraction for the production of grape juice, a new product, according to our knowledge, is proposed as an alternative for the valorization of this fruit. The effect of the extraction time on the composition and antioxidant capacity of País grape juice obtained was evaluated, as well as the change in the phenolic profile during storage. The soluble solid values and total polyphenol and total anthocyanin content increased with the extraction time. However, a residence time of the juice in the extraction device higher than 10 min led to thermal degradation of anthocyanins and flavonols. The most abundant phenolic compound identified and quantified by HPLC-DAD in the País grape juice was cinnamic acid. The storage of juices had a greater effect on anthocyanin and flavonol losses than the residence time of the juice in the extraction device. The antioxidant capacity of juice measured by ABTS and ferric reducing power assays ranged from 3 to 5 mmol trolox/L and from 10 to 18 mM $Fe^{2+}$/L, respectively. In summary, steam extraction is a viable method to produce País grape juice with antioxidant capacity.

**Keywords:** país grape; juice; steam extraction; phenolic compounds; antioxidant capacity

## 1. Introduction

The economic development of Chilean agriculture is linked to viticulture; the quality of Chilean wines has opened the world to Chilean agricultural products. The first strains of *Vitis vinifera* L. were introduced by Spanish monks during colonial times [1]. One of the pioneer strains was *V vinifera* L. cv. País, also known as Mission or Criolla. Currently, this red grape variety is the fourth most abundant planted in the country at around 10,319 ha [2]. However, the País grape is not a favorite for wine producers because its wine is characterized by harshness and imbalance [3]. These undesirable oenological characteristics have led to its low market price, insufficient to cover the production costs of smallholder family farming. Thus, part of the seasonal grapes are not harvested, and grapes are lost in the vineyard. The challenge is to use this undervalued and ancestral grape in the making of other valuable products.

Grape juices are a rich source of polyphenol compounds [4], and their consumption contributes to human health [5]. The absence of alcohol makes the juice an excellent product for mass consumption, particularly for children, the elderly, and pregnant women. The use of the País variety for juice production has been documented by Aguilar et al. [6].

Polyphenol-enriched grape juice was prepared by thermo-maceration of a mixture of grape must and pomace for 6 h. The authors reported a high antioxidant capacity of the juice of 55 mmol equivalent of trolox/L juice. However, the maceration method is carried out between one h to seven days, which favors the appearance of mycotoxins, such as ochratoxin A, in the presence of spoiled grapes with toxigenic fungi [7].

An alternative method for grape juice production with reduced mycotoxin and increased polyphenol content is steam extraction. The thermal method uses steam to break down the cell membranes and extract the juice from the fruit [8]. Steam extraction is used industrially on small and medium scales to facilitate extraction and extend the shelf life of fruit juice without the need for separate pasteurization. In addition, steam displaces the oxygen present, favoring the stability of polyphenolic compounds such as anthocyanins during the extraction. Lopes et al. [9] found that the steam extraction method produced grape juice with a higher anthocyanin content and higher antioxidant capacity and stability than juices obtained by (cold) mechanical treatments.

The steam extraction time is a relevant parameter in juice production due to the instability of polyphenolic compounds at high temperatures [10]. However, temperature and extraction time can also increase the juice polyphenol content [7,11,12]. It is necessary to find a good balance between the extraction time and phenolic profile in País grape juice produced by the steam extraction method. The novelty of this work is the preparation of juice from País grapes, a new product according to our knowledge, by the steam extraction method. The effect of the extraction time on the composition and antioxidant capacity of País grape juice obtained was evaluated, as well as the change in the phenolic profile during storage.

## 2. Materials and Methods

### 2.1. Materials

#### 2.1.1. Grape Samples

The grapes were harvested from an estate located in the commune of San Nicolás ($36°32'22.2''$ S and $72°20'08.8''$ W) in the Punilla Province, Ñuble Region, Chile, in the 2020–2021 season. The average total soluble solid (SS) content of the processed grapes was 25.15 °Brix due to the advanced maturity stage of the berries. For this analysis, seven berries were randomly selected for each replicate. The the commune of San Nicolás is characterized by a warm temperate climate with winter rainfall. Annual precipitation fluctuates between 800 and 1100 mm. The soils are of granitic type with textures ranging from clay to sandy loam.

#### 2.1.2. Chemical Reagents

The analytical standards used in liquid chromatography assay were petunidin-3-glucoside (≥95.0%, Supelco, Bellefonte, PA, USA), peonidin-3-glucoside (≥97%, Sigma–Aldrich, Burlington, MA, USA), malvidin 3-glucoside (≥95.0%, Merck, Kenilworth, NJ, USA), quercetin-3-glucoside (≥98%, Supelco, Bellefonte, PA, USA), quercetin dihydrate (≥98%, Merck, Kenilworth, NJ, USA), myricetin (≥98%, Merck, Kenilworth, NJ, USA), gallic acid (≥98.0%, Merck, Kenilworth, NJ, USA), chlorogenic acid (≥95.0%, Merck, Kenilworth, NJ, USA), 4-hydroxybenzoic acid (≥99%, Merck, Kenilworth, NJ, USA), caffeic acid (≥95.0%, Merck, Kenilworth, NJ, USA), *p*-coumaric acid (≥98.0%, Sigma Aldrich, Burlington, MA, USA), benzoic acid (≥98.0%, Sigma Aldrich, Burlington, MA, USA), and cinnamic acid (≥98.0%, Merck, Kenilworth, NJ, USA). Ultrapure water, LiChorsolve, was supplied by Merck. Folin–Ciocalteu's reagent and gallic acid provided by Merck, Darmstalt, Germany were used in the evaluation of the total polyphenol content. 2,2′-azino-bis(3-ethylbenzothiazoline-6-sulphonic acid) (ABTS) diammonium salt (Sigma Aldrich, St. Louis, MO, USA), 6-hydroxy-2,5,7,8-tetramethyl chroman-2-carboxylic acid (Trolox) (EMD Chemicals, San Diego, CA, USA), and 2,4,6-tris(2-pyridyl)-s-triazine (TPTZ) (Sigma Aldrich, St. Louis, MO, USA) were used for antioxidant capacity assays.

*2.2. Methods*

2.2.1. Preparation of Grape Juice

The grape juice was obtained using a domestic steam juicer (HOMESTYLE Comfort, Oyten, Germany). The grapes were charged in the upper fruit basket and exposed to a constant flow of water vapor (>80 °C) rising from the lower part of the equipment, causing the softening and rupture of the fruit cell walls and subsequent leaching of the juice [7]. The dripping juice from the juice basket was collected and pasteurized in an annular middle part that was heated by the rising steam. The juice residence time depended on the juice production rate and the emptying of the juice receiver by the operator. The juices were hot filled in glass containers of 250 mL. The water: fruit ratio was 1:3 v/wt. The juice yield was approximately 2 L of juice/4 kg of grapes. Sampling of the produced juice started at 10 min, 20 min or 30 min. Then, the extraction process was followed by taking additional samples of approximately 500 mL every 10 min. The receiver was emptied completely after each sample extraction. The samples were stored at 15 °C for up to 60 days to evaluate their shelf life during storage. The juices were also analyzed at 0, 30, and 60 days. For comparative purposes, juice was produced by mechanical extraction using a Hurom slow juicer (Hurom LS Co, Ltd., Gyeongsangnam-do, Korea). No other treatment was applied to this juice to prolong its shelf-life. The sample identification used, (TN)x, specifies the time N of the run at which sampling started and x when the sample was taken. The time was measured from the first appearance of juice in the equipment outlet valve. Thus, for example, $T10_{30}$ was a juice fraction collected in the run where sampling started at 10 min. The fraction was collected at time 30 min and corresponded to the juice produced in 10 min and accumulated in the equipment between 20 and 30 min. Therefore, samples $T10_{10}$, $T10_{20}$, and $T10_{30}$ follow the extraction process every 10 min, while $T10_{10}$, $T20_{20}$, and $T30_{30}$, reflect the cumulative production and storage at high temperature for 10, 20, and 30 min, respectively.

2.2.2. Soluble Solids, pH, and Titratable Acid Assays for Juice Samples

The classic parameters of juice, pH (potentiometer pH LAQUA PH 1200); the soluble solids (°Brix, refractometer RHB-32ATC, Hilab, Amsterdam, The Netherlands); and the titratable acidity, were determined according to the methodology described in reference [13].

2.2.3. Total Phenolic Content of Juice Samples

The total phenolic content (TPC) was estimated by the Folin–Ciocalteu assay as previously described [14]. In short, a sample or blank of 200 μL was added to 1000 μL of Folin–Ciocalteu's reagent (1:10, *v:v*) followed by 800 μL $Na_2CO_3$ solution (7.5% wt/v). The juice samples were previously diluted to 1:26 *v:v*. Samples were heated for 15 min at 45 °C. The absorbance was determined at 765 nm with a Shimadzu UV 2600 spectrophotometer (Shimadzu corporation, Kyoto, Japan). The TPC was reported as gallic acid equivalents (mg GAE) per L of grape juice. The calibration curve was constructed with gallic acid in the concentration range of 0–10 mg/L. The TPC measurement of the juices was done in triplicate.

2.2.4. Total Anthocyanin Content of Juice Samples Measured by UV Spectrometry

The total anthocyanin content was measured spectrophotometrically (Shimadzu UV 2600 Spectrophotometer, Shimadzu corporation kyoto, Japan), using the pH differential method with cyanidin-3-O-glucoside (c3-Og) as a standard [15]. Two absorbances were read at 520 and 700 nm at different pH values. Two aliquots of 200 μL of juice sample were separately diluted in sodium acetate buffer (0.4 M, pH = 4.5) and in potassium chloride buffer (0.025 M, pH = 1). The quantity of anthocyanin was calculated using the following equation: A × MW × DF × 1000)/ε, where A is absorbance measured at different values of λ (520 or 700 nm) and pH, MW, ε molecular weight, and the molar extinction coefficient

for c3-Og (26.900 L cm$^{-1}$ mol$^{-1}$), and DF is the dilution factor. All samples were measured in triplicate.

### 2.2.5. Identification and Quantification of Phenolic Compounds of Juice Samples by HPLC–DAD

The phenolic compounds were extracted (1:1) by volume with a methanol:formic acid:water (25:1:24) solution as described by Gironés-Vilaplana et al. [16]. The HPLC analyses were performed using an Primade Hitachi high performance liquid chromatograph (Merck, Darmstadt, Germany) coupled to a diode array detector (1430 Primade) and a Kromasil 100-5-C18 column (250 mm × 4.6 mm, 5 μm particle size; Nouryon, Separation Products, Stockholm, Sweden), with 1% formic acid and acetonitrile as mobile phases A and B, respectively. The flow rate was 1 mL·min$^{-1}$. The injection volume of the juice sample was 10 μL, which was previously filtered with a PVDF syringe filter with a 0.22 μm pore size. The identification of the most abundant phenolic compounds was achieved by combining the retention time and spectral characteristic of each analytical standard (petunidin-3-glucoside, peonidin-3-glucoside, malvidin 3-glucoside, quercetin dihydrate, myricetin, gallic acid, chlorogenic acid, 4-hydroxybenzoic acid, caffeic acid, *p*-coumaric acid, benzoic acid, and cinnamic acid). The procedure for phenolic compound identification was validated according to the reference [17]. The quantification of the phenolic compounds was based on calibration curves of analytical standards, and the results were expressed in mg L$^{-1}$ of juice. For anthocyanins, the calibration curve of cyanidin 3-O-glucoside was constructed at 520 nm in the concentration range of 0.5 to 100 ppm (y = 8874x + 853, R$^2$ = 0.9986). Quantification of malvidin derivatives was done according to the calibration graphs of malvidin-3-O-glucoside. For flavonols, the calibration curve of quercetin dihydrate was constructed at 360 nm in the range of concentration 0.1–10 ppm (y = 45,790x − 424, R$^2$ = 0.9974). For hydroxycinnamic acids, the calibration curve of caffeic acid was measured at 320 nm in the concentration range 0.1–10 ppm (y = 62,322x − 2155, R$^2$ = 0.9981). For hydroxybenzoic acids, the calibration curve of 4-hydroxybenzoic acid was constructed at 280 nm in the range of concentration 0.1–10 ppm (y = 51,490x − 5052, R$^2$ = 0.9976). The limits of detection and quantification were 0.10 and 0.31, respectively.

### 2.2.6. Antioxidant Capacity Based on Ferric-Ion Reducing Antioxidant Power (FRAP)

The in vitro antioxidant capacity of juices was determined using the FRAP assay reported in [14] with some modifications. The FRAP reagent was made by combining acetate buffer (0.3 mol L$^{-1}$, pH = 3.6), TPTZ solution dissolved in HCl 40 mmol L$^{-1}$, and 20 mmol·L$^{-1}$ FeCl$_3$ in a ratio of 10:1:1. A FeSO$_4$ aqueous solution in the range of 10 to 50 μmol/L was used for the calibration curve. Then, 100 μL of diluted juice sample (dilution factor = 20) and 3 mL of fresh FRAP reagent were mixed by vortexing. The solution was incubated in a digital triple heat block (model 12621-110, VWR, Radnor, PA, USA) at 37 °C for 30 min. The absorbance of the working standards versus the control was recorded at 593 nm. FRAP values were reported as mg FeSO$_4$ per L grape juice. The samples were measured in triplicate.

### 2.2.7. Antioxidant Capacity Based on the Radical Cation Assay (ABTS$^{\bullet+}$)

ABTS diammonium salt (ABTS$^{\bullet+}$) cation radical scavenging assay was performed following [18,19] with some modifications. The ABTS$^{\bullet+}$ radical cations stock solution was prepared by reaction of 2.45 mmol L$^{-1}$ of potassium persulphate and 7 mmol L$^{-1}$ of ABTS aqueous solution. The mixture was kept in the dark at room temperature for 16 h. The ABTS$^{\bullet+}$ solution was then diluted with absolute ethanol and allowed to stand for at least 6 h until an absorbance of 0.7 at 734 nm was reached. The calibration curve was prepared from 0.5 mmol L$^{-1}$ of stock solution of Trolox, the antioxidant standard in this method, in absolute ethanol. The concentration range of the Trolox working solutions was 2.27 to 14 mmol L$^{-1}$. Absolute ethanol was used as a reference. A 100 μL of juice sample was diluted in absolute ethanol (dilution factor = 50). A 100 μL of the sample supernatant, previously diluted with a dilution factor = 2 and centrifuged at 10,000 rpm for 10 min, or

Trolox working solutions, was mixed by vortexing with 1 mL of diluted ABTS$^{\bullet+}$ solution before absorbance measurements. The absorbance reading at 734 nm was taken exactly 6 min after mixing ($A_{6min}$) and before adding the sample or Trolox solutions at ABTS$^{\bullet+}$ ($A_0$). The free radical scavenging activity was calculated as follows: ABTS scavenging activity (%) = [($A_0 - A_{6min}$)/$A_0$] × 100. The percentage of ABTS scavenging activity at 734 nm was plotted as a function of the concentration of the Trolox standards. Samples were analyzed in triplicate. The antioxidant capacity was expressed as µmol of Trolox per L of grape juice.

### 2.2.8. Statistical Analysis

The statistical treatment of experimental data was performed using GraphPad Prism 9 software (GraphPad Software Inc., San Diego, CA, USA). The data were analyzed by one-way ANOVA, and $p < 0.05$ was considered statistically significant. In the case of parametric data, Student's test was applied for comparison of two groups and Tukey's test for three or more groups. For nonparametric data, the Kruskal–Wallis and Dunn's tests were applied.

## 3. Results and Discussion

### 3.1. Soluble Solids, pH, and Titratable Acid Analysis

Soluble solids (SS), pH, and titratable acid (TA) reflect the juice quality, due to their influence on the organoleptic properties of the juice. The values of these classic quality parameters for the juice samples obtained at different extraction times are summarized in Table 1. The pH value and TA are also used as indicators of microbiological alterations. The SS values are in accordance with those recommended by the Chilean legislation for fruit juice [20], which establishes a minimum of 20%wt soluble solids of the ripe fruit from which it is derived.

**Table 1.** The quality parameters of grape juices obtained by steam and mechanical extraction processes.

| Grape Juice Sample [1] | SS (°Brix) | pH | TA (g Tartaric Acid/L) | SS/TA |
|---|---|---|---|---|
| T10$_{10}$ | 14 ± 3 [a] | 3.82 ± 0.05 [a] | 2.9 ± 0.2 [a] | 5 ± 1 [a] |
| T10$_{20}$ | 18.5 ± 0.5 [b] | 3.83 ± 0.02 [a,b] | 3.0 ± 0.1 [a] | 6.2 ± 0.1 [a,b] |
| T10$_{30}$ | 20.0 ± 0.7 [b,c] | 4.0 ± 0.1 [a,b] | 2.7 ± 0.2 [a] | 7.3 ± 0.8 [a,b] |
| T20$_{20}$ | 16.1 ± 0.6 [a,b,d] | 3.78 ± 0.02 [a] | 2.72 ± 0.03 [a] | 5.9 ± 0.4 [a,b] |
| T20$_{30}$ | 17.7 ± 0.1 [a,b,c] | 3.80 ± 0.01 [a] | 2.96 ± 0.07 [a] | 6.0 ± 0.2 [a] |
| T30$_{30}$ | 18.1 ± 0.3 [b,c,d] | 3.9 ± 0.2 [a,b] | 2.9 ± 0.1 [a] | 6.3 ± 0.3 [a,b] |
| Extruded | 26.4 ± 0.3 [e] | 4.1 ± 0.2 [b] | 3.0 ± 0.8 [a] | 9 ± 2 [b] |

[1] For explanation of sample codes, see Section 2.2.1. Values followed by different lowercase letters (a, b, c, d, e) within the same column are significantly different at $p < 0.05$ based on the Tukey test.

The SS of the grape juices was found to increase with the extraction time. Thus, the SS value of T10$_{30}$ (20.0 °Brix) was significantly higher than that of T10$_{10}$ (14.0 °Brix). At the beginning of the extraction process, the dilution of the juice was the greatest because of the condensation of steam on the cold equipment walls and fruits. However, as juice production increased (not shown), no significant dilution effect was observed when comparing T10$_{30}$ and T30$_{30}$, despite that the former only included juice produced between 20 and 30 min and the latter included all the juice produced from the beginning. The mean value for the SS in the T10$_{30}$ juices is in agreement with results reported by Lima et al. [21] for grape juice prepared from Isabel Precoce and BRS Violeta varieties (20.3 °Brix) grown in a similar condition to País grapes. The SS values of the juice obtained by the mechanical method (26.4 °Brix) were similar to the SS values of the fruits used in the extractions and were higher than the values obtained by steam extraction due to dilution by condensed steam and soluble solids trapped in the extraction residue. In contrast, the values of TA and pH did not appreciably change for grapes exposed to the mechanical or steam extraction processes. The dilution of the juice by condensed steam is partially compensated by the

extraction of additional acids from the grape skins [22]. Unfortunately, Chilean legislation, unlike that of Brazil, does not specify pH and TA values for grape juices. The pH and TA of juices are grape cultivar-dependent [22]. In general, TA values of País grape juices are lower than those obtained for Brazilian varieties using different extraction methods: steam, mechanical, and hot maceration [23,24], but higher than those of muscadine grape juices [25].

The mean values of the SS/TA ratio were between 5 and 9 for juices obtained by both mechanical and steam extraction. The ratio denotes the balance between the sweet and acid taste in fruit juices. It is important to note that no differences were found in the values of SS, pH, and TA of grape juice obtained by the steam process at the same extraction time during 60 days of storage (data not shown).

### 3.2. Total Phenolics and Anthocyanin Content Based on the Spectrophotometric Method

Total phenolics and total anthocyanins in the País grape juice obtained by steam extraction at different initial extraction times and different storage times are shown in Figures 1 and 2, respectively. The mechanical procedure was only evaluated at time zero due to the lack of pasteurization. The total phenolic content (TPC) of País grape juices was found to be in a wide range from 790 to 1490 mg of GAE/L. As expected, the juice sample $T10_{10}$ showed a lower TPC than that of $T10_{30}$, and this trend was maintained throughout the 60 days of storage. For T20 juice samples, no significant differences were found between samples collected at 20 and 30 min. Furthermore, the TPC values of the T10 and T20 juice samples did not differ statistically from those of the juice obtained by mechanical extraction. This occurred despite the fact that during mechanical processing, a disruption of the grape matrix occurs, resulting in a higher extraction of phenolic compounds from the seeds, which have the highest levels of phenolic compounds [26]. However, the TPC values of sample $T30_{30}$ were significantly lower than those of mechanically extracted juice. The steam applied during extraction contributes to increase the extractability of phenolic compounds [24], but a high residence time in the extraction device and steam temperature may have had a detrimental effect on the TPC of $T30_{30}$ juice.

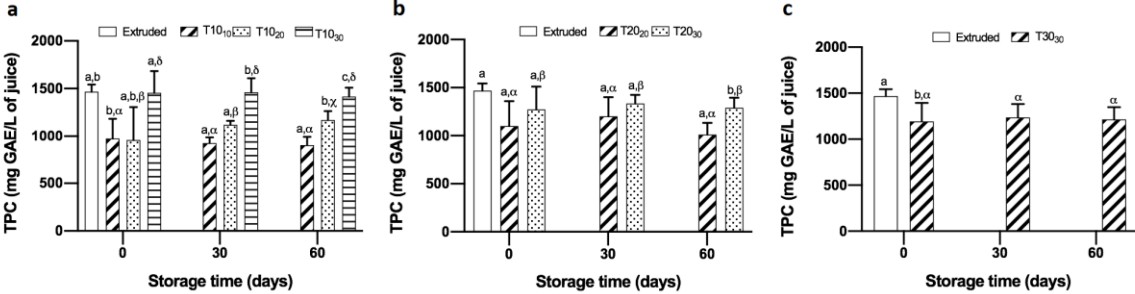

**Figure 1.** TPC of País grape juices at different initial extraction times determined by the Folin–Ciocalteu method, before and after 30 and 60 days of storage, where (**a**) is for juice samples collected at 10 min of the run at which sampling started ($T10_{10}$, $T10_{20}$, $T10_{30}$); (**b**) is for juice samples collected at 20 min of the run at which sampling started ($T20_{20}$, $T20_{30}$) and (**c**) is for juice samples collected at 30 min of the run at which sampling started ($T30_{30}$). The columns with different lowercase letters for the same storage times are significantly different at $p < 0.05$ based on Tukey's test in the case of the T10 samples and Student's test in the case of the T20 and T30 samples. The columns with different symbols ($\alpha$, $\beta$, $\delta$, $\chi$) for the same initial extraction time and different storage times are significantly different at $p < 0.05$ based on Tukey's test.

In general, the TPC values of País grape juices reported by us, except for sample $T10_{10}$, are in agreement with those found by Moreno-Montero et al. [27] and da Silva et al. [23] for commercial red grape juices (790 to 1774 mg GAE $L^{-1}$), but were lower than those found by Lopes et al. [9] for grape juice of *Vitis labrusca* cv. Isabel produced by the steam extraction method ($2727 \pm 94$ mg GAE $L^{-1}$). The content of phenolic compounds is, among other factors, grape cultivar-dependent [28].

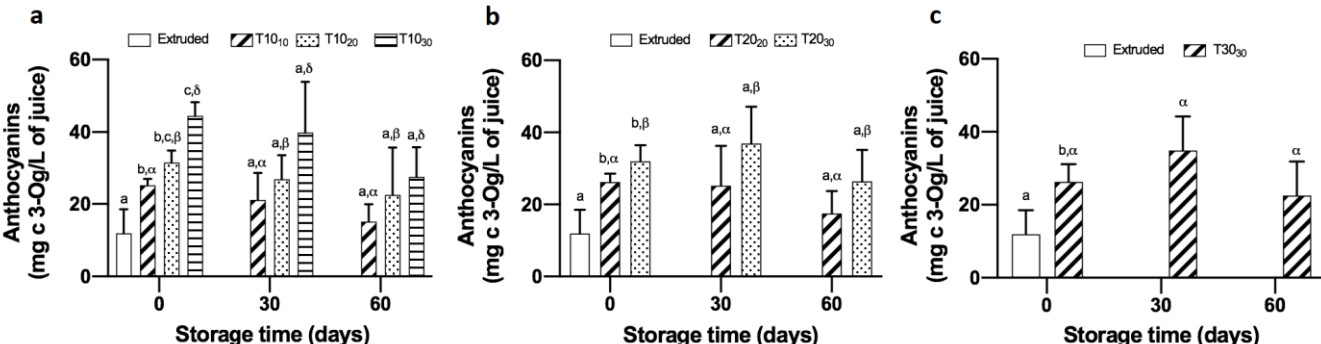

**Figure 2.** Total anthocyanin content of País grape juices at different initial extraction times determined by pH differential assay, before and after 30 and 60 days of storage, where (**a**) is for juice samples collected at 10 min of the run at which sampling started (T10$_{10}$, T10$_{20}$, T10$_{30}$); (**b**) is for juice samples collected at 20 min of the run at which sampling started (T20$_{20}$, T20$_{30}$) and (**c**) is for juice samples collect-ed at 30 min of the run at which sampling started (T30$_{30}$). The columns with different lowercase letters for the same storage time are significantly different at $p < 0.05$ based on Tukey's test in the case of the T10 samples and Student's test in the case of the T20 and T30 samples. The columns with different symbols ($\alpha$, $\beta$, $\delta$) for the same initial extraction time and different storage times are significantly different at $p < 0.05$ based on Tukey's test.

During storage, no appreciable changes were observed in the TPC values of the País grape juices, although this is not indicative of the stability of specific phenolic compounds. In particular, anthocyanins are susceptible to oxidation and their contents decline during storage [29].

The total anthocyanin contents of grape juice obtained by steam extraction and quantified by the pH differential method ranged from 23.4 to 47.1 mg L$^{-1}$ on day 0. The T10$_{30}$ juice samples (44.5 mg L$^{-1}$) had the highest total anthocyanin content, and the juice prepared by the mechanical procedure the lowest values (11.9 mg L$^{-1}$), which is consistent with the results reported by Lopes et al. [9]. Heat applied during the process increased the extractability of anthocyanins. For T10 samples (Figure 1a), with a maximum juice residence time of 10 min at high temperature, a significant increase in the anthocyanin content with increasing extraction time was observed (25.2, 31.6 and 44.5 mg L$^{-1}$ for T10$_{10}$, T10$_{20}$, and T10$_{30}$, respectively). For the T20 sample (Figure 2b), no significant changes in the anthocyanin content were found between samples T20$_{20}$ (26.2 mg L$^{-1}$) and T20$_{30}$ (32.0 mg L$^{-1}$). The anthocyanin content of the juices exposed to high temperature for 30 min (26.3 mg L$^{-1}$) was similar to that of the T10$_{10}$ sample, showing the detrimental effect of temperature on the anthocyanin content of juices accumulating in the equipment for more than 20 min

The total anthocyanin contents of all juice samples, regardless of extraction time, decreased during storage, although the differences were not statistically significant. As in the case of wine aging, a reduction in the anthocyanin content in the juice is expected during storage due the reduction of monomeric anthocyanins and formation of polymeric pigments. Condensation reactions between phenolic compounds and anthocyanins may also reduce the anthocyanin content in the juice during storage [30,31].

### 3.3. Identification and Quantification of Phenolic Compounds of Juice Samples by HPLC–DAD

The main phenolic compounds of the fruit juice identified and quantified by HPLC-DAD are listed in Table 2, and their chemical structures are presented in Figure 3. In general, the polyphenol content obtained spectrophotometrically was higher than that found by HPLC, which is associated to the ability of the Folin–Ciocalteu mixture to react with any reducing substance present in the juice matrix. On the other hand, in the HPLC-DAD analysis, only the most abundant secondary metabolites in the juice were quantified (flavonoids, and hydroxybenzoic and hydroxycinnamic acids) [4], but these are not the only polyphenols in the juice [27].

**Table 2.** Phenolic compounds (mg/L) in grape juices at different initial extraction times, measured by HPLC, before and after 30 and 60 days of storage.

| Storage Time (Days) | $T10_{10}$ | $T10_{20}$ | $T10_{30}$ | $T20_{20}$ | $T20_{30}$ | $T30_{30}$ | Extruded |
|---|---|---|---|---|---|---|---|
| **Anthocyanins** | | | | | | | |
| Petunidin-3-glucoside　0 | $0.63 \pm 0.03$ [a,c,α] | $0.72 \pm 0.07$ [a,b,α] | $0.76 \pm 0.03$ [a,b,α] | $0.54 \pm 0.02$ [c,α] | $0.53 \pm 0.04$ [c,α] | $0.52 \pm 0.04$ [c,α] | $0.76 \pm 0.1$ [a,b] |
| 　30 | $0.04 \pm 0.01$ [a,β] | $0.201 \pm 0.003$ [b,β] | $0.16 \pm 0.02$ [c,β] | $0.15 \pm 0.01$ [c,β] | $0.17 \pm 0.01$ [c,β] | $0.16 \pm 0.01$ [c,β] | |
| 　60 | $0.15 \pm 0.02$ [a,γ] | $0.13 \pm 0.02$ [a,β] | $0.116 \pm 0.005$ [a,b,γ] | $0.10 \pm 0.01$ [b,c,γ] | $0.136 \pm 0.006$ [a,b,β] | $0.13 \pm 0.03$ [a,b,c,β] | |
| Peonidin-3-glucoside　0 | ND | ND | $0.027 \pm 0.001$ [a,α] | ND | ND | ND | $0.168 \pm 0.002$ [b] |
| 　30 | ND | $0.032 \pm 0.007$ [a] | $0.036 \pm 0.008$ [a,α] | ND | ND | $0.024 \pm 0.007$ [a] | |
| 　60 | ND | ND | ND | ND | ND | ND | |
| Malvidin 3-glucoside　0 | $0.26 \pm 0.01$ [a,α] | $0.313 \pm 0.003$ [b,α] | $0.41 \pm 0.01$ [c,α] | $0.24 \pm 0.01$ [a,α] | $0.27 \pm 0.01$ [a,α] | $0.24 \pm 0.01$ [a,α] | $0.44 \pm 0.03$ [c] |
| 　30 | N.D. | $0.10 \pm 0.03$ [a,β] | $0.08 \pm 0.02$ [a,b,β] | $0.05 \pm 0.02$ [a,b,β] | $0.042 \pm 0.004$ [b,β] | $0.05 \pm 0.01$ [a,b,β] | |
| 　60 | $0.07 \pm 0.01$ [a,β] | $0.09 \pm 0.01$ [a,c,β] | $0.03 \pm 0.02$ [b,γ] | $0.06 \pm 0.01$ [a,β] | $0.10 \pm 0.01$ [c,γ] | $0.07 \pm 0.02$ [a,c,β] | |
| Malvidin derivative　0 | $2.01 \pm 0.03$ [a,c,d,α] | $2.08 \pm 0.09$ [a,d,α] | $2.6 \pm 0.3$ [b,d,α] | $1.6 \pm 0.2$ [a,c,α] | $1.4 \pm 0.2$ [c,α] | $1.7 \pm 0.2$ [a,c,α] | $3.1 \pm 0.4$ [b] |
| 　30 | $0.5 \pm 0.1$ [a,b,β] | $0.72 \pm 0.05$ [b,β] | $0.38 \pm 0.07$ [a,c,β] | $0.37 \pm 0.01$ [a,e,β] | $0.05 \pm 0.01$ [d,β] | $0.29 \pm 0.02$ [c,e,β] | |
| 　60 | $0.14 \pm 0.02$ [a,γ] | $0.24 \pm 0.03$ [b,γ] | $0.24 \pm 0.01$ [b,β] | $0.21 \pm 0.03$ [b,c,β] | $0.24 \pm 0.03$ [b,γ] | $1.8 \pm 0.01$ [c,β] | |
| Total anthocyanins　0 | 2.90 | 3.12 | 3.77 | 2.40 | 2.17 | 2.47 | 4.47 |
| 　30 | 0.58 | 1.06 | 0.65 | 0.57 | 0.26 | 0.52 | |
| 　60 | 0.35 | 0.46 | 0.38 | 0.38 | 0.48 | 0.38 | |
| **Flavonols** | | | | | | | |
| Quercetin-3-glucoside　0 | $0.51 \pm 0.02$ [a,α] | $0.61 \pm 0.05$ [a,α] | $1.8 \pm 0.2$ [b,α] | $0.68 \pm 0.07$ [a,c,α] | $0.9 \pm 0.1$ [c,d,α] | $1.02 \pm 0.05$ [d,α] | $1.46 \pm 0.03$ [e] |
| 　30 | $0.30 \pm 0.01$ [a,β] | $0.54 \pm 0.04$ [b,α] | $1.28 \pm 0.05$ [c,β] | $0.74 \pm 0.01$ [d,α] | $1.46 \pm 0.02$ [e,β] | $0.69 \pm 0.03$ [e,β] | |
| 　60 | $0.36 \pm 0.02$ [a,γ] | $0.41 \pm 0.02$ [a,b,β] | $0.52 \pm 0.11$ [b,γ] | $0.43 \pm 0.8$ [a,b,β] | $0.45 \pm 0.06$ [a,b,γ] | $0.45 \pm 0.09$ [b,γ] | |
| Myricetin 3-O-glucoside　0 | $0.76 \pm 0.02$ [a,α] | $1.08 \pm 0.09$ [b,α] | $1.4 \pm 0.2$ [c,α] | $0.48 \pm 0.02$ [d,α] | $0.72 \pm 0.08$ [a,e,α] | $0.99 \pm 0.03$ [b,α] | $0.71 \pm 0.04$ [a] |
| 　30 | $0.48 \pm 0.02$ [a,β] | $1.05 \pm 0.03$ [b,α] | $1.20 \pm 0.06$ [c,β] | $0.74 \pm 0.01$ [d,β] | $0.77 \pm 0.08$ [d,α] | $1.03 \pm 0.04$ [b,α] | |
| 　60 | $0.35 \pm 0.09$ [a,β] | $0.49 \pm 0.01$ [b,β] | $0.82 \pm 0.05$ [c,γ] | $0.29 \pm 0.05$ [a,γ] | $0.53 \pm 0.07$ [b,d,β] | $0.53 \pm 0.08$ [a,b,d,β] | |
| Total flavonols　0 | 1.28 | 1.69 | 3.22 | 1.16 | 1.58 | 2.01 | 2.17 |
| 　30 | 0.777 | 1.59 | 2.48 | 1.48 | 2.23 | 1.72 | |
| 　60 | 0.716 | 0.90 | 1.35 | 0.72 | 0.98 | 0.90 | |
| **Hydroxybenzoic (HB) and Hydroxycinnamic (HC) Acids** | | | | | | | |
| Gallic acid　0 | $0.51 \pm 0.05$ [a,α] | $0.8 \pm 0.1$ [b,c,α] | $1.0 \pm 0.1$ [c,α] | $0.67 \pm 0.03$ [a,b,α] | $0.77 \pm 0.08$ [b,c,α] | $0.8 \pm 0.2$ [b,c,α] | $0.80 \pm 0.04$ [b,c] |
| 　30 | $0.42 \pm 0.02$ [a,β] | $0.87 \pm 0.04$ [b,α] | $1.0 \pm 0.1$ [c,α] | $0.63 \pm 0.05$ [d,α] | $1.27 \pm 0.06$ [e,β] | $1.11 \pm 0.04$ [c,α] | |
| 　60 | $0.58 \pm 0.03$ [a,γ] | $1.3 \pm 0.2$ [b,β] | $1.4 \pm 0.3$ [b,α] | $0.77 \pm 0.09$ [a,c,α] | $1.0 \pm 0.1$ [c,d,α] | $1.1 \pm 0.2$ [b,d,α] | |

**Table 2.** *Cont.*

| | Storage Time (Days) | $T10_{10}$ | $T10_{20}$ | $T10_{30}$ | $T20_{20}$ | $T20_{30}$ | $T30_{30}$ | Extruded |
|---|---|---|---|---|---|---|---|---|
| Chlorogenic acid | 0 | $3.1 \pm 0.3$ [a,b,α] | $3.04 \pm 0.03$ [a,b,α] | $3.32 \pm 0.02$ [a,b,α] | $2.2 \pm 0.1$ [a,α] | $2.10 \pm 0.01$ [a,b,α] | $2.66 \pm 0.01$ [a,α] | $4.22 \pm 0.02$ [b] |
| | 30 | $1.96 \pm 0.01$ [a,β] | $2.72 \pm 0.08$ [a,b,β] | $2.71 \pm 0.01$ [a,b, β] | $2.22 \pm 0.01$ [a,b,α] | $2.8 \pm 0.1$ [b,β] | $2.69 \pm 0.01$ [a,b,α] | |
| | 60 | $2.8 \pm 0.2$ [a,α] | $3.0 \pm 0.2$ [a,b,α] | $3.48 \pm 0.09$ [b,c,γ] | $2.71 \pm 0.07$ [a,β] | $2.8 \pm 0.2$ [a,β] | $2.9 \pm 0.2$ [a,c,α] | |
| 4-Hydroxybenzoic acid | 0 | $0.61 \pm 0.0$ [a,α] | $0.88 \pm 0.08$ [b,α] | $2.6 \pm 0.1$ [c,α] | $0.81 \pm 0.08$ [b,α] | $1.84 \pm 0.05$ [d,α] | $1.55 \pm 0.03$ [e,α] | $3.94 \pm 0.02$ [f] |
| | 30 | $0.55 \pm 0.04$ [a,α] | $0.9 \pm 0.1$ [b,α] | $3.3 \pm 0.2$ [c,β] | $1.48 \pm 0.01$ [d,β] | $1.9 \pm 0.1$ [e,α] | $2.15 \pm 0.08$ [e,β] | |
| | 60 | $1.04 \pm 0.03$ [a,c,β] | $1.07 \pm 0.06$ [a,α,β] | $1.20 \pm 0.04$ [b,γ] | $0.94 \pm 0.03$ [d,γ] | $0.99 \pm 0.03$ [c,d,β] | $1.00 \pm 0.06$ [a,d,γ] | |
| Caffeic acid | 0 | $5.43 \pm 0.05$ [a,α] | $6.0 \pm 0.1$ [b,α] | $6.16 \pm 0.08$ [b,c,d,α] | $4.80 \pm 0.01$ [c,α] | $4.7 \pm 0.1$ [c,α] | $5.71 \pm 0.06$ [d,α] | $5.89 \pm 0.05$ [b,d] |
| | 30 | $3.71 \pm 0.01$ [a,β] | $5.9 \pm 0.1$ [b,α] | $6.8 \pm 0.3$ [c,β] | $4.99 \pm 0.01$ [d,α] | $6.6 \pm 0.5$ [c,e,β] | $6.15 \pm 0.01$ [b,e,β] | |
| | 60 | $5.0 \pm 0.2$ [a,γ] | $5.8 \pm 0.3$ [b,α] | $6.7 \pm 0.1$ [c,β] | $5.1 \pm 0.2$ [a,α] | $5.9 \pm 0.1$ [b,γ] | $5.94 \pm 0.09$ [b,γ] | |
| *p*-Coumaric acid | 0 | $0.84 \pm 0.01$ [a,α] | $0.96 \pm 0.01$ [b,α] | $1.12 \pm 0.01$ [c,α] | $0.71 \pm 0.02$ [d,α] | $0.73 \pm 0.01$ [d,α] | $0.86 \pm 0.01$ [a,α] | $1.53 \pm 0.01$ [f] |
| | 30 | $0.55 \pm 0.01$ [a,β] | $0.92 \pm 0.01$ [b,β] | $0.99 \pm 0.01$ [c,β] | $0.72 \pm 0.01$ [d,α] | $1.46 \pm 0.05$ [e,β] | $0.98 \pm 0.01$ [c,α] | |
| | 60 | $0.78 \pm 0.04$ [a,α] | $0.90 \pm 0.1$ [b,β] | $1.17 \pm 0.05$ [c,α] | $0.82 \pm 0.02$ [a,d,β] | $0.95 \pm 0.02$ [b,γ] | $0.89 \pm 0.08$ [b,d,α] | |
| Benzoic acid | 0 | $1.0 \pm 0.2$ [a,α] | $1.45 \pm 0.06$ [b,α] | $1.8 \pm 0.1$ [c,α] | $1.46 \pm 0.05$ [b,α] | $1.54 \pm 0.08$ [b,α] | $1.50 \pm 0.01$ [b,α] | $2.56 \pm 0.03$ [d] |
| | 30 | $0.79 \pm 0.02$ [a,α,β] | $1.35 \pm 0.04$ [b,α] | $2.20 \pm 0.08$ [c,β] | $1.28 \pm 0.01$ [b,α] | $2.3 \pm 0.1$ [c,β] | $1.91 \pm 0.05$ [b,α] | |
| | 60 | $1.1 \pm 0.2$ [a,β] | $1.4 \pm 0.2$ [a,b,α] | $1.7 \pm 0.2$ [b,c,α] | $1.3 \pm 0.1$ [a,d,α] | $1.89 \pm 0.06$ [c,γ] | $1.6 \pm 0.3$ [b,c,d,α] | |
| Cinnamic acid | 0 | $39.5 \pm 0.4$ [a,α] | $39.4 \pm 0.8$ [a,α] | $41.2 \pm 0.1$ [b,α] | $29.6 \pm 0.2$ [c,α] | $28.4 \pm 0.2$ [d,α] | $35.2 \pm 0.3$ [e,α] | $44.3 \pm 0.2$ [f] |
| | 30 | $25.8 \pm 0.1$ [a,β] | $38.0 \pm 0.1$ [b,c,β] | $40.6 \pm 0.1$ [b,d,α] | $33.09 \pm 0.02$ [a,c,d,β] | $43 \pm 2$ [b,β] | $37.58 \pm 0.01$ [a,d,β] | |
| | 60 | $35.4 \pm 0.7$ [a,γ] | $38.6 \pm 0.1$ [b,β] | $41.7 \pm 0.8$ [c,α] | $33 \pm 1$ [d,β] | $36 \pm 1$ [a,γ] | $38.4 \pm 0.2$ [b,γ] | |
| Total HB and HC acids | 0 | 50.9 | 52.5 | 57.2 | 40.2 | 40.2 | 48.2 | 63.2 |
| | 30 | 33.8 | 50.6 | 57.7 | 44.4 | 59.4 | 52.6 | |
| | 60 | 46.7 | 52.1 | 57.3 | 45.4 | 50.01 | 51.9 | |
| **Total Phenolic Compounds** | | | | | | | | |
| | 0 | 55.1 | 57.3 | 64.2 | 43.8 | 43.9 | 52.7 | 69.8 |
| | 30 | 35.1 | 53.3 | 60.8 | 46.5 | 61.8 | 54.8 | |
| | 60 | 49.0 | 55.0 | 61.1 | 47.8 | 53.1 | 53.2 | |

The results are expressed as the mean $\pm$ standard deviation (n = 3). The columns with different symbols (α, β, γ) for the same phenolic compound and different storage times are significantly different at $p < 0.05$ based on Tukey's test and Student's test for two groups. The rows with different lowercase letters (a, b, c, d, e, f) for the same phenolic compound and different extraction times are significantly different at $p < 0.05$ based on Tukey's test (parametric) and the Kruskal-Wallis and Dunn's tests (nonparametric, Chlorogenic acid). ND: not detected.

**Figure 3.** Chemical structures of the phenolic compounds detected by HPLC-DAD in País grape juice obtained by steam extraction.

The maldivin derivative was the most abundant anthocyanin in the juices obtained by stream extraction, accounting for ~70% of the total anthocyanin content, followed by petunidin-3-glucoside (~20%) and malvidin glucoside (~10%). Peonidin-3-glucoside was identified in some juice samples at a concentration lower than 0.03 mg L$^{-1}$. The sum of the individual anthocyanins quantified by HPLC ranged from 2.17 to 3.77 mg L$^{-1}$. The extruded juice showed a similar profile before storage; however, the anthocyanin content was higher, around 15%. The T10$_{30}$ juice sample had the highest total anthocyanin content of the steam-extracted samples, which agrees with the results obtained by the pH differential assay. In the late stage of steam extraction, mainly skin components are extracted [4]. The extruded juice showed a similar profile, and without any heat treatment, the anthocyanin content was about 15% higher. After the storage of juices for 60 days at 15 °C, the concentration of the quantified anthocyanins decreased by one order of magnitude. As mentioned above, the monomeric anthocyanins are highly unstable and susceptible to degradation at room temperature [32].

Quercetin-3-glucoside and myricetin 3-O-glucoside were flavonols identified in the juices obtained by steam and mechanical processes. The sum of the individual flavonols quantified by HPLC ranged from 1.16 to 3.22 mg/L, and their content depended on the extraction time. Similar profiles of flavonols were identified in juice obtained by the mechanical procedure. Aguilar et al. [6] reported other glycosylated flavonols such as myricetin, kaempferol, and isorhamnetin in País grape juice enriched with cane and leaf grape extracts. The T10$_{30}$ sample, regardless of the storage time, had a higher total flavonol content compared with other juice samples. The level of flavonols decreased between

37–58% after 60 days of storage. This loss of flavonols may be related to their oxidation to quinones or their adsorption on insoluble solids, which precipitate late during storage [33]. In general, the storage time of grape juices at 15 °C had a greater effect on anthocyanin and flavonol losses than the extraction time and high temperature residence time used in the juice production.

The four hydroxycinnamic acids found in the grape juice were cinnamic acid, chlorogenic acid, caffeic acid, and *p*-coumaric acid. The three hydroxybenzoic acids were 4-hydroxybenzoic acid, benzoic acid, and gallic acid. The most abundant type of acids found in the grape juice were cinnamic acids, representing around 90% of the total acids. The sum of the individual hydroxycinnamic and hydroxybenzoic acids quantified by HPLC ranged from 40.2 to 57.2 mg/L. In contrast to flavonols and anthocyanins, the levels of hydroxybenzoic and hydroxycinnamic acids were stable during the 60 days of storage.

### 3.4. Antioxidant Capacity of the País Grape Juice Obtained by Steam Extraction

The antioxidant capacity of País grape juices obtained by steam extraction was determined by two different methods, FRAP and ABTS (Table 3). The methods for the determination of the antioxidant capacity are divided into two main groups: assays based on the one-electron transfer reaction and assays based on the transfer of a hydrogen atom. The electron transfer method (FRAP) evaluates the ability of a potential antioxidant to transfer an electron to reduce a metal compound. The proton transfer method (ABTS) measures the ability of an antioxidant to trap free radicals by donating a hydrogen. However, electron transfer is also involved in the ABTS assay. Before storage, the antioxidant capacity of País grape juices obtained by steam extraction and estimated by FRAP and ABTS assays did not differ significantly for samples obtained throughout the extraction process (T10$_{30}$, T20$_{30}$, and T30$_{30}$), despite the differences in the levels of phenolic compounds between samples, mainly in anthocyanins and flavonols. In addition, the antioxidant capacity values were low compared to those reported by Aguilar et al. [6] for polyphenol-enriched juices of País grape (55 mmol equivalent of trolox/L juice). The antioxidant capacity measured by ABTS assay decreased by around 30% after 60 days of storage regardless the extraction time, which correlated with the losses of anthocyanins and flavonols. The results of the FRAP assay were not clearly related to a change in the phenolic profile after 60 days of storage. The antioxidant capacity of the T20$_{30}$ and T30$_{30}$ samples increased significantly (by ~16%) after storage, while the T10$_{30}$ sample did not show a significant change.

**Table 3.** Antioxidant capacity of juice obtained by steam extraction at different initial extraction times and before and after 60 days of storage.

| Storage Time (Days) | T10$_{30}$ | | T20$_{30}$ | | T30$_{30}$ | |
|---|---|---|---|---|---|---|
| | ABTS | FRAP | ABTS | FRAP | ABTS | FRAP |
| 0 | 5.9 ± 0.9 [a,α] | 17 ± 5 [a,α] | 5 ± 1 [a,α] | 12.0 ± 0.5 [a,α] | 5.3 ± 0.4 [a,α] | 10 ± 2 [a,α] |
| 60 | 3.8 ± 0.4 [a,β] | 18.0 ± 0.8 [a,α] | 3.02 ± 0.02 [b,β] | 14.0 ± 0.3 [b,β] | 3.0 ± 0.1 [b,β] | 13.3 ± 0.1 [a,β] |

Values of ABTS are expressed as mM Trolox/L of juice and values of FRAP as mM $Fe^{2+}$/L of juice. The rows with different lowercase letters (a, b) for the same antioxidant assay are significantly different at $p < 0.05$ based on Student's test. The columns with different symbols (α, β) for the same initial extraction time and different storage times are significantly different at $p < 0.05$ based on Student's test.

## 4. Conclusions

The small-to-medium scale production of grape juice from *V vinifera* L. cv. País by means of steam extraction allowed us to obtain a stable juice rich in polyphenols with an antioxidant capacity that does not require any further physical or chemical treatment for at least 60 days of storage. The organoleptic and microbiological indicators represented by the SS content, pH, and TA remained stable during the storage period. Soluble solids and the total phenolic and anthocyanin contents increased with the extraction time, and only the anthocyanin content clearly decreased with increasing residence time of the juice in the hot receiver, which functions as a pasteurization stage. A strong decrease in

the anthocyanins identified and quantified by HLPC-DAD during storage could not be detected by spectrophotometric analysis of total anthocyanins.

**Author Contributions:** Conceptualization, S.R.-L. and G.P.; methodology, W.I., C.S. and N.P.; validation, W.I.; formal analysis, W.I., R.B., L.M. and A.L.; investigation, W.I.; resources, S.R.-L.; data curation, W.I.; writing—original draft preparation, S.R.-L.; writing—review and editing, S.R.-L.; visualization, J.C.; supervision, N.M. and S.R.-L.; project administration, S.R.-L.; funding acquisition, S.R.-L. All authors have read and agreed to the published version of the manuscript.

**Funding:** This research was funded by Fortalecimiento de Centros Regionales Mediante Proyectos de I+D Ciencia–Territorio, ANID, Chile, grant number R18F10016, and Apoyo a la Formación de Redes Internacionales entre Centros de Investigación, ANID, Chile grant number REDES190181.

**Data Availability Statement:** The datasets generated for this study are available on request to the corresponding author.

**Conflicts of Interest:** The authors declare no conflict of interest.

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
