# Peer review of "Grape (Vitis vinifera L. cv. País) Juices Obtained by Steam Extraction"

_processes, doi:10.3390/pr9091670_

Round 1
Reviewer 1 Report
Article is interesting, but must be corrected. Methodological errors and text editing are required:
1) What is the novelty of the research in relation to those published in the literature? Improve the justification in introduction and abstract.
2) On methodology: Specify in detail how compounds were quantified by HPLC-DAD. Has the method been validated? If possible, specify the purity of the compounds.
3) Describe in detail how the compounds were identified by HPLC-DAD-ESI-MSn, showing possible transitions and fragmentations.
4) Draw the chemical structures of the compounds.
5) Standardize the paragraphs according to the journal's standards.
Reviewer 2 Report
The manuscript titled “Grape (Vitis vinifera L. cv. País) juices obtained by steam-extraction: Phenolic compound profile, antioxidant activity and time stability” is using technology that is not novel, the only novelty is utilization of Pais variety. English should be polished by a native speaker as some sentences are unclear and hard to understand. There are also some issues listed below which should be addressed.
Page 1 Lines 23-24 – Please add missing text in the sentence with TP and antiox assays
Page 1 Lines 41-42 – Please rephrase the sentence. Furthermore, reference should be numbered appropriately.
Page 1 Lines 41-46 – If Pais variety is abundant why it is not harvested? This doesn’t make a lot of sense. Please rephrase it.
Page 2 Lines 52-54 – Reference should be numbered
Page 3 Section 2.2.1. There were three initial times 10, 20 and 30min. What was happening during that time, grapes were in the extractor with steam? Please add more information?
Page 3 Line 119 – 15 C is not aligned with information from abstract, please correct it where needed
Please add section Statistical analysis after 2.2.7. and briefly explain used statistical tools.
Table 1 – lowercase letters within SS column are not appropriate, Extruded should be marked with the “a” and T1010 with “e” (or the other way around). Please perform statistical analysis also for SS/TA and provide standard deviation.
Page 5 Lines 225-226 – Lima et al. (17) – Does not exist in the reference list. Furthermore, ref 17 is Pellegrini et al. Please check the entire reference list.
Page 5 Lines 228-229 – Which juice extractions? Generally speaking?
Page 5 Lines 241-243 – storage (data not shown).
Figures 1 and 2 – Please explain why extruded grape juice was not pasteurized so it could be analyzed during storage time, and thus compared to steam extraction samples?
Page 6 Lines 259-261 – The sentence is incomplete, please correct it.
Page 6 Lines 267-268 – The highest value for TPC of any sample was ~1500 mg GAE/L, how is this in agreement with ~2700 mg GAE/L? Please rephrase the sentence.
Section 3.3. Since HPLC–DAD–ESI-MSn was used, were there some other phenolic compounds that were identified (but not quantified)? Please add this additional phenolic in a separate table and present it as a phenolic profile. Also, add one chromatogram as figure.
Page 8 Line 322 - Aguilar et al.
Page 8 Line 331 – This sentence is too short, add more information.
Table 4. Why antioxidant activity was not analyzed after 30 days of storage?
Typos and English
Page 1 Line 25 – 25 ºC
Page 3 Line 124 – content from juice which was
Page 6 Line 253 and Page 8 line 309 – zero
Page 8 Line 314 – lixiviate is maybe not the best term
Round 2
Reviewer 1 Report
Authors provided the corrections. Thus, this paper can be published after removing strikethrough texts.
Author Response
Thanks to the reviewer for the comments. The reference list was revised again and some typographical errors were corrected. The abstract was modified to reach a maximum of 200 words. In addition, the strikethrough texts were removed in second revise manuscript.
Reviewer 2 Report
Authors improved the manuscript, however there are still issue with references.
Page 7 Line 354 - There is still Lima et al. [19] and in the reference list numb 19 is Re, Roberta, Nicoletta Pellegrini, Anna Proteggente, Ananth Pannala, Min Yang, and Catherine Rice-Evans. "Antioxidant Activity Applying an Improved Abts Radical Cation Decolorization Assay." Free Radical Biology and Medicine 26, no. 9 (1999):1231-37. The entire reference list should be checked again. Once corrected it could be accepted.
Author Response
Thanks to the reviewer for the comments. The reference list was checked again and some typographical errors were corrected. The abstract was modified to reach a maximum of 200 words.